# Pharmacological Optimization of PSMA-Based Radioligand Therapy

**DOI:** 10.3390/biomedicines10123020

**Published:** 2022-11-23

**Authors:** Suzanne van der Gaag, Imke H. Bartelink, André N. Vis, George L. Burchell, Daniela E. Oprea-Lager, Harry Hendrikse

**Affiliations:** 1Department of Radiology and Nuclear Medicine, Amsterdam UMC Location Vrije Universiteit Amsterdam, De Boelelaan 1117, 1081 HV Amsterdam, The Netherlands; 2Cancer Center Amsterdam, Imaging and Biomarkers, De Boelelaan 1118, 1081 HV Amsterdam, The Netherlands; 3Department of Clinical Pharmacology and Pharmacy, Amsterdam UMC Location Vrije Universiteit Amsterdam, De Boelelaan 1117, 1081 HV Amsterdam, The Netherlands; 4Department of Urology, Prostate Cancer Network Amsterdam, Amsterdam UMC Location Vrije Universiteit Amsterdam, De Boelelaan 1117, 1081 HV Amsterdam, The Netherlands; 5Medical Library, VU University, De Boelelaan 1117, 1081 HV Amsterdam, The Netherlands

**Keywords:** PSMA, prostate cancer, radioligand, therapy, theranostics, optimization, pharmacokinetics, pharmacodynamics, variability, lutetium

## Abstract

Prostate cancer (PCa) is the most common malignancy in men of middle and older age. The standard treatment strategy for PCa ranges from active surveillance in low-grade, localized PCa to radical prostatectomy, external beam radiation therapy, hormonal treatment and chemotherapy. Recently, the use of prostate-specific membrane antigen (PSMA)-targeted radioligand therapy (RLT) for metastatic castration-resistant PCa has been approved. PSMA is predominantly, but not exclusively, expressed on PCa cells. Because of its high expression in PCa, PSMA is a promising target for diagnostics and therapy. To understand the currently used RLT, knowledge about pharmacokinetics (PK) and pharmacodynamics (PD) of the PSMA ligand and the PSMA protein itself is crucial. PK and PD properties of the ligand and its target determine the duration and extent of the effect. Knowledge on the concentration–time profile, the target affinity and target abundance may help to predict the effect of RLT. Increased specific binding of radioligands to PSMA on PCa cells may be associated with better treatment response, where nonspecific binding may increase the risk of toxicity in healthy organs. Optimization of the radioligand, as well as synergistic effects of concomitant agents and an improved dosing strategy, may lead to more individualized treatment and better overall survival.

## 1. Introduction

Prostate cancer (PCa) is the most common malignancy in men of middle and older age worldwide, with an incidence of 14.1% of all cancers [1]. The age-standardized mortality rate lies between 3.1 and 27.9 per 100,000 people. High-income countries have relatively low mortality rates due to more advanced treatment options [1]. Treatment strategies for PCa range from active surveillance in low-grade, localized PCa to radical prostatectomy, external beam radiation therapy, hormonal treatment and chemotherapy. Recently, the United States Food & Drug Administration (FDA) has approved the use of prostate-specific membrane antigen (PSMA)-targeted radioligand therapy (RLT) for metastatic castration-resistant PCa [2].

PSMA is a transmembrane protein of approximately 100 kDa, also called glutamate carboxypeptidase II, N-acetyl-L-aspartyl-L-glutamate peptidase I or folate hydrolase [3]. PSMA is involved in multiple physiological processes. The most important processes are glutamate signal transduction by PSMA in the brain, nutrient uptake from glutamates by PSMA in the GI-tract and a receptor function in PCa cells [4]. PSMA is predominantly, but not exclusively, expressed on PCa cells [3,4]. Due to its function, PSMA is expressed on several healthy tissues, e.g., normal prostate tissue, small intestine, central nerve system, proximal renal tubules and salivary glands. PSMA is significantly overexpressed in most PCa cells (>90%) and its expression is even higher in high-grade, aggressive and dedifferentiated tumors [5]. In addition, PSMA is also expressed in other types of malignancies, among which are pancreatic, colorectal, gastric, central nerve system and renal cell carcinoma [6,7,8,9]. Because of its high expression in PCa, PSMA is a promising target for diagnostics and therapy (so called theranostics). Theranostics is an approach for using a similar molecule for diagnostics as well as for targeted therapy, creating an individualized treatment option. An example of theranostics in the PCa setting is the use of positron-emitter ^68^Ga-PSMA for diagnostics and a beta- (^177^Lu) or alpha-emitter (^225^Ac) coupled to PSMA for RLT. Positive outcomes for PSMA RLT have been demonstrated [10].

To understand the current use of PSMA-RLT, knowledge about pharmacokinetics (PK) and pharmacodynamics (PD) of the PSMA ligand and the PSMA protein itself is crucial. PK and PD properties of both the ligand and its target determine the duration and extent of the effect. Knowledge of the concentration–time profile, the target affinity and target abundance may help to predict the effect of RLT. Increased specific binding of radioligands to PSMA on PCa cells may be associated with better treatment response, where nonspecific binding may increase the risk of toxicity in healthy organs.

For PSMA-based RLT, monoclonal antibodies and small molecules have been developed. Due to their molecular size, monoclonal antibodies have a relatively poor permeability into solid tumors and a slow blood clearance [11]. Therefore, monoclonal antibodies are less preferred for RLT when compared to small molecules. Small molecules targeting PSMA, such as PSMA-617 and PSMA-I&T, have a larger permeability in solid tumors and a faster blood clearance, which leads to higher tumor uptake and higher specific binding to PSMA receptors [11]. PSMA-617 and PSMA-I&T have similar pharmacokinetic, safety and efficacy profiles and are thus comparable among clinical studies [12,13,14,15]. Due to the favorable properties of small-molecule PSMA-based radioligands, the focus of this review is on small molecules. 

Prior studies on PSMA-RLT mainly focused on clinical outcomes and adverse events, and knowledge is limited on PK/PD of RLT. To our knowledge, this is the first structured review on PK and its relationship with duration, early (biomarkers of) response, long-term outcomes and toxicity. Therefore, in this article we provide a structured overview on PK and PD properties and their influence on various aspects of RLT. In addition, we propose a pharmacologically based optimization strategy for PSMA RLT and thus promising options for future research.

## 2. Methods

A systematic search was performed in the following databases: PubMed, Embase.com, Clarivate Analytics/Web of Science Core Collection and the Wiley/Cochrane Library. The timeframe within the databases was from inception to 17 October 2022 and was conducted by GLB and SG. The search included keywords and free text terms for (synonyms of) ‘prostate specific membrane antigen’ combined with (synonyms of) ‘Randomized Controlled Trial’. A full overview of the search terms per database can be found in the Appendix A. No limitations on language were applied in the search.

Duplicate articles were excluded using the R-package “ASYSD”, an automated deduplication tool [16], followed by manual deduplication in Endnote (X20.0.3) by the medical information specialist (GLB).

## 3. Pharmacokinetics of PSMA-Based Radioligand Therapy

Pharmacokinetics describes the fate of the radioligand in the body and includes drug distribution, metabolism and elimination.

### 3.1. Distribution

PSMA radioligands are hydrophilic compounds (log D of PSMA-617 is −4.4), causing a rapid distribution throughout the body and a large volume of distribution [17]. Kratochwil et al., described a volume of distribution 1 h after injection with ^177^Lu-PSMA-617 of 22 ± 12 L, which is comparable to the amount of extracellular volume [18]. Since PSMA radioligands have no strong affinity for blood components such as albumin or specific transporters, the main route of distribution to PSMA-negative cells is by passive diffusion over the cell membrane into the interstitial space of the organs [18,19,20]. However, due to their hydrophilic nature, which is not favorable for passive diffusion, most of the PSMA radioligand amount remains in the bloodstream. PSMA-expressing cells have receptors on the cell surface that capture and subsequently internalize the PSMA radioligands, causing accumulation into the cytoplasm of the cell [18,19,20]. Approximately 75% of all radioligands with affinity for PSMA are internalized within the first 3 h after incubation on LNCaP cells [18]. This rapid intracellular uptake of ^177^Lu-PSMA-617 and ^177^Lu-PSMA-I&T is also described in humans, where a maximum uptake of both ligands is found between 4 and 24 h after injection [21,22]. A long-term retention in tumor cells and metastases was observed with a time to maximum standardized uptake value (SUV_max_) of up to 118 h after injection [23]. ^177^Lu-PSMA-617 distributes to organs that express PSMA, such as kidneys and salivary glands [22]. Tumors with a high volume are more likely to have a higher number of PSMA receptors and a significantly higher radioligand uptake compared to low-volume tumors [24]. This study concluded that tumor volume was predictive for PSMA radioligand tumor distribution, while difference in this parameter did not affect the drug (bio)distribution to the organs-at-risk [24].

### 3.2. Elimination (Metabolism and Excretion)

Metabolism by the liver is only relevant for agents with a more lipophilic character. PSMA-617 and PSMA-I&T have multiple hydroxyl groups and subsequently have pKa values of approximately four. This pKa value means that in blood, at pH 7, the molecules are ionized. Due to their hydrophilicity and ionization in blood, PSMA radioligands are suitable for renal excretion. Firstly, the compounds undergo glomerular filtration, where the molecular weight is an important characteristic. Both ^177^Lu-PSMA-617 and ^177^Lu-PSMA-I&T have a molecular weight of 1216.1 and 1498.4 Da, respectively [25,26]. As molecules below 20,000 Da are well suited for glomerular filtration, PSMA ligands are almost completely excreted from the body by glomerular filtration [23,27]. Inside the nephron, there is no active secretion or tubular reabsorption for these small molecules as they are not lipophilic. In addition, Kratochwil et al., estimated the fecal excretion to be between 1 and 5% of the total clearance [18,28]. Renal excretion of unbound PSMA radioligands can be enhanced by diuretics and the small portion of fecal excretion can be enhanced by [29].

Clearance of PSMA radioligands is biexponential, with a fast clearance from blood and non-PSMA-expressing organs around 2–4 h and a significant slower clearance from the PSMA-expressing organs around 41–160 h (*p* = 0.001) [11,18,30,31,32,33]. PSMA-based radioligands have an effective half-life between 21 and 91 h [18,23,24]. The organs-at-risk have a shorter half-life compared to tumor tissue, namely 33 ± 14 and 51 ± 30 h, respectively [11,18,30,31,32,33]. PSMA radioligands have a fast clearance from non-PSMA-expressing organs since they have no target binding and therefore are not internalized and do not accumulate in the cell. This results in a higher mean drug exposure in the tumor compared to healthy tissues. Pharmacokinetics are linear among dose levels as no correlation between injected dose and measured half-life was observed [28].

Summarizing, the rapid tissue distribution to and accumulation in PSMA-expressing organs and rapid clearance of the remaining radioligands from PSMA-negative organs leads to favorable characteristics for imaging and therapy of PSMA radioligands. It is worth mentioning that not all radionuclides have the same concentration–time profile. After decay to their respective daughter nuclides, some emitters may show a different distribution profile. For example, the alpha-emitter ^225^Ac has a significant redistribution of daughter nuclides from the decay chain of ^225^Ac throughout the body [34].

### 3.3. Variability in Pharmacokinetics

Pharmacokinetic modeling has shown a large interindividual variability in radioligand exposure to tumors and organs-at-risk, which can be predicted for patient, drug and tissue characteristics [35,36,37,38]. Weight and estimated glomerular filtration rate (eGFR) did not influence tumor-PK, but tumor volume associated positively with radioligand exposure in the tumor tissue [36]. These results were confirmed by Begum et al., who identified that a tumor-sink effect influenced radioligand exposure [37]. Patients with a high tumor load may be prone to benefit from this tumor-sink effect, as the consequence of the tumor-sink effect is that healthy tissues show decreased uptake. In addition to patient characteristics, drug- and tissue-specific properties such as ligand amount, affinity and internalization rate are associated with radioligand exposure [38]. At microdose levels used for diagnostics, low amounts of activity, predictable distribution and a lack of tumor-sink effect limited the assessment of PK variability among patients [37]. However, receptor density and organ flow can be accurately estimated by positron emission tomography (PET) at a microdose level, before the start of the first therapy cycle [39]. RLT PK analyses showed that variability in patient- and drug-specific variables influenced drug tumor uptake [35,37,38]. A higher tumor radioligand exposure correlated with a higher radioligand exposure in the kidneys [37]. Higher affinity for the receptor, higher internalization and higher receptor density led to higher radioligand exposure in tumor tissue [35,38]. These studies suggest that measurements of receptor density and organ flow based on the diagnostic scan can be useful to individualize RLT treatment. In case of high organ flow or high receptor density, a higher ligand dose may increase the radioligand exposure of the tumor and therefore may benefit RLT treatment. 

Summarizing, variability in PK of PSMA-based radioligands is most likely caused by tumor volume, receptor density and organ blood flow. Individualization of therapy can be established by pretreatment PET with measurement of these patient-specific parameters.

### 3.4. Optimizing Pharmacokinetic Properties of the Radioligand

Several studies have been conducted to further optimize the pharmacokinetic properties of PSMA radioligands. One strategy is to increase the binding affinity to albumin to prolong residence time of the drug in blood. Albumin-binding affinity can be increased by adding an albumin-binding motif or adapting linker [3,17,40,41,42,43,44,45,46,47]. 

The addition of an albumin-binding motif resulted in an increased residence time of the ligand in blood [17,41,45,46,48,49]. The increased residence time in blood prolongs exposure of the ligand in the proximity of the target receptor, and thus increases the possibility for the ligand to bind to the receptor and be internalized. The extent of reversibility of binding to albumin influenced the degree of availability of the ligand, as only the unbound ligand is available for receptor binding. Two studies observed a 2.8- and 14-fold higher accumulation in PSMA-expressing tissues with two small molecules, including an albumin-binding motif [41,48]. The favorable properties of the Evans blue (EB) albumin-binding motif currently seems to be the best option [46,47]. EB is an albumin-binding motif with moderate binding affinity, and therefore reversible binding to albumin. First-in-human studies showed promising results with a significant higher tumor uptake, confirming the results in cell lines that showed an increased uptake of 77.3 ± 6.2% at 48 h compared to PSMA radioligands without the EB-binding motif (*p* < 0.01) [46,47]. The disadvantage of a longer residence time is the relatively high renal uptake, and thus increased risk of accumulation and renal damage [46,47]. Unfortunately, a 6-fold higher kidney uptake compared to PSMA-617 (*p* < 0.001) was observed by Zang et al. [46,47]. In a dose escalation study with EB-PSMA-617, this increased kidney uptake did not significantly influence the renal functions of—nor the treatment response to—^177^Lu-EB-PSMA within the 2 months postdose observation period [47].

Another strategy for optimizing the pharmacokinetic properties is adaptions in the linker moieties between the pharmacore and the DOTA chelator [3,40,43,50]. These adaptations can further optimize the positioning of the ligand in the binding pocket of PSMA and thus create a better affinity for the PSMA receptor, as investigated by Kopka et al. [43]. These adaptations also create a more lipophilic molecule, causing the lowering of renal clearance, followed by a longer residence time in blood and kidneys, and in turn causing toxicity to the organs-at-risk [43]. Ultimately, a balance needs to be found in increasing tumor accumulation by adapting protein-binding properties without increasing the risk of toxicity by a prolonged residence time.

## 4. Pharmacodynamics of PSMA-Based Radioligand Therapy

### 4.1. PSMA Receptor as a Target

The PSMA receptor consists of 750 amino acids, where 19 are located intracellularly, 24 in the transmembrane region and 707 extracellularly [3,43]. A schematic overview of the receptor is shown in Figure 1. The intracellular domain has a physiological function contributing to the extraction of nutrients from ingested food and mediating the internalization of PSMA ligands [4,51]. After binding of the PSMA ligand to PSMA, the complex is internalized into the cell by clathrin-mediated endocytosis through clathrin-coated pits and transported into endosomes [52]. These endosomes fuse with lysosomes. After fusion, the PSMA is recycled and transported back to the plasma membrane or degraded by lysosomal degradation [52]. The ligand, however, is released in the cell and can exert its effect. 

The extracellular part of PSMA consists of three subdomains, namely the protease, the apical and the helical domain [3]. Within these three domains, a large cavity arises, which forms the substrate-binding site of the PSMA receptor. Inside the binding complex, an active site with two zinc ions—the pharmacore—with a chelator and a glutamate-sensing pocket, as well as an arene-binding site, can be distinguished [3,43,50]. The two zinc ions interact with the negatively charged carboxyl groups of the PSMA ligand. The most suitable amino acids for this interaction are urea and glutamate, where glutamate also fits perfectly into the glutamate-sensing pocket [3]. The DOTA chelator, used in PSMA-617 and PSMA-I&T, is the most appropriate chelator for arrangement inside the PSMA-binding domain of the receptor compared to other chelators, including HBED-CC and DOTAGA-FFK [3,43,50]. Furthermore, the hydrophobic interactions with the lipophilic, nonpolar domain of the PSMA ligand are provided by the arene-binding site [3,48,49]. All these interactions in the binding pocket of the PSMA receptor lead to a favorable affinity (e.g., 2.0 ± 0.3 nM for ^177^Lu-PSMA-617) [53].

### 4.2. Predictors of Response

Prostate-specific antigen (PSA) is used as a surrogate marker for tumor volume in PCa patients. PSA is known to be a strong prognostic factor but does not always accurately determine treatment outcomes [54,55]. Nevertheless, PSA and its decline are widely used biomarkers of treatment response, as there are objective and relatively easily obtainable measurements. However, clinical accuracy needs to be improved [54,55]. In a review, PSMA receptors and bone metabolic markers, such as N-telopeptide, bone-specific alkaline phosphatase, C-telopeptide and pyridinoline, predicted treatment response to RLT, whereas a change in PSA was associated with treatment efficacy of androgen-receptor inhibitors (ARIs) only [56]. In another study of 14 castration-resistant PCa patients, baseline PSMA activity was a strong predictor for treatment response to RLT, defined as a PSA decline of >30% [57]. The correlation between PSMA expression and PSA secretion and whether they are regulated by the same mechanism is currently unknown [58,59].

Manafi et al., performed a comprehensive review on factors that predict the biochemical response and survival to ^177^Lu-PSMA RLT. Initial PSA and PSA doubling time did not correlate with progression-free survival (PFS) or overall survival (OS) after treatment onset, where baseline PSA before treatment has a controversial correlation with survival as studies are contradictory in positive and negative correlation with survival [60]. Assessment of PSA decline after treatment should be combined with functional PET imaging, as any decrease in PSA led to a significantly better OS (*p* < 0.001), but an increase in PSA did not lead to a worse OS [60]. In some patients, no change in PSA levels from baseline was observed, but imaging showed radiographic disease progression [60]. Finally, visceral, bone and lymph node metastases were predictive of negative treatment outcomes, possibly based on a high severity of the disease [60,61].

Summarizing, to accurately predict and evaluate treatment response, functional PET imaging is the best objective biomarker. As functional PET imaging is time-consuming and not readily available in all treatment facilities, PSA as a surrogate marker can be used to monitor treatment efficacy. Note that PSA is not the most optimal predictive biomarker, but the best available biomarker we have today.

### 4.3. Synergistic Effects

#### 4.3.1. Synergy by Upregulation of PSMA

PSMA protein expression is indirectly regulated by androgens. In the normal physiological situation, androgens, such as testosterone, enter the cell by passive diffusion through the cell membrane [58]. Testosterone is intracellularly converted into dihydrotestosterone (DHT), forming a complex with the androgen receptor (AR). After forming the DHT-AR complex, the PSMA enhancer region, promoting gene transcription, is suppressed, leading to an inactivation of the folate hydrolase 1 (FOLH1) gene, and thus blockage of the synthesis of the PSMA protein (Figure 2a) [58]. Synthesis of the PSMA protein might be up- and downregulated by drugs. ARIs such as enzalutamide and bicalutamide competitively inhibit the formation of a DHT-AR complex, and thereby inhibit the inactivation of FOLH1 [62,63,64,65,66]. This leads to activation of the enhancer region, and as a result, more synthesis of the PSMA protein (Figure 2b).

This upregulation pathway caused by ARIs has been well described in human PCa cell lines. LNCaP and C4-2 cell lines were exposed to 0.25–10 µM enzalutamide, abiraterone, bicalutamide and dutasteride during the period of 72 h to 14 days [62,63,64,65,66,67]. The cellular upregulation of PSMA after exposure to these drugs was 1.9–7.4-fold, 1.5–5-fold, 4.2-fold and 2.4–4.6-fold, respectively [62,63,64,65,66,67]. The upregulation appeared to be concentration-dependent with a certain threshold and was more predominant in castration-resistant cells compared to hormone-sensitive cells. This predominant upregulation is most likely the result of more PSMA-expressing cells in castration-resistant PCa [67].

Before the effect of radioligands can be achieved in PCa patients, the drugs need to be internalized into the cell. A dose-independent increase in internalization of PSMA radioligands and subsequent increase in ^177^Lu-PSMA-617 internalization was observed after adding enzalutamide and dutasteride to LNCaP cells [64,65].

In vivo, a 1.9-fold upregulation of PSMA in tumor-bearing mice (C4-2 xenograft) by 10 mg/kg once daily (QD) apalutamide for 7 days was observed [68,69]. Furthermore, a 4.2-fold upregulation of PSMA in 20 tumor-bearing mice (LNCaP xenograft) followed 23 days with 10 mg/kg enzalutamide or 50 mg/kg bicalutamide QD, assessed by flow-cytometric quantification of tumor biopsies at 4 h, 48 h and 96 h post-RLT [69]. In this study, increased PSMA expression led to significantly increased DNA damage by ^177^Lu-PSMA-617 and significant improved survival from 34 (range 19–62) days to 81 (range 62–81) days (*p* < 0.001) in 20 mice using this combination therapy compared to 20 mice using vehicle [69]. These data suggest a promising strategy towards upregulation of PSMA by ARIs in humans.

A clinical case report describing upregulation of PSMA by ARIs observed a 7-fold higher PSMA-uptake, measured as SUV_max_ on a post-treatment ^68^Ga-PSMA-11 PET, after treatment with luteinizing hormone-releasing hormone (LHRH) agonists: a single dose of 7.5 mg leuprolide followed by 50 mg bicalutamide QD for a 30-day period [68]. Another patient with a low-PSMA-expressing tumor showed an increased PSMA expression after 5 months of 160 mg enzalutamide treatment QD from 4.4 to 16.4, measured by SUV on ^68^Ga-PSMA-11 PET/CT [67]. Subsequently, Emmett et al., performed a study including 15 patients, divided into 2 cohorts. Cohort 1 included 8 patients with hormone-sensitive PCa treated with 50 mg bicalutamide and a LHRH-agonist and cohort 2 included 7 patients with castration-resistant PCa treated with enzalutamide or abiraterone and a LHRH-agonist [70]. Cohort 2, as expected, showed 45% upregulation of PSMA on day 9 (IQR 12.7-66). The increase plateaued at days 18 to 28. Cohort 1, however, showed a 30% reduction in PSMA expression on day 9 (IQR 5-61), with a further reduction on day 18 and 28 [70]. The authors speculated that upregulation by enzalutamide is present in hormone-sensitive PCa but is overruled by the treatment response of hormone-naïve PCa cells to ARIs creating tumor shrinkage, and thus a lesser extent of PSMA expression. In none of these patients, salivary glands showed a significant change in PSMA expression from baseline [70]. In 10 CRPC patients treated with 160 mg enzalutamide QD for 11.8 ± 8.1 days, a 45–55% higher PSMA uptake was observed compared to untreated patients (*p* = 0.014) [59]. Treatment with ARIs in an earlier stage of the disease did not have a significant impact on the upregulation of PSMA by ARIs in a later stage [59,71]. Similar results were found in a retrospective analysis of pre- and post-enzalutamide ^68^Ga-PSMA-11 PET scans, with an increase of 49.6% in SUV_max_ in target lesions after 13 ± 7 days of enzalutamide treatment [71]. A randomized phase II trial comparing enzalutamide alone with enzalutamide plus ^177^Lu-PSMA-617 in patients with mCRPC, the so-called ENZA-p trial, is still pending [72]. The extent of PSMA upregulation may be time-dependent, with optimal expression after 9–14 days of treatment. Kranzbuhler et al., described a tendency for downregulation after 14 days of treatment of LNCaP cells with 5 µM dutasteride [65]. In this study, PSMA levels returned to baseline approximately seven days after omitting ARIs [65]. A downregulation of PSMA by active vitamin D has been observed, but it is still unclear what the mechanism is and whether downregulation by vitamin D correlates with an upregulation of the AR protein [73].

Summarizing, these preclinical and clinical findings show that upregulation of PSMA may be accomplished by ARIs. The mechanism behind this upregulation is still not fully elucidated, as ARIs were combined with LHRH agonists in the above-mentioned studies, making the effect of monotherapy with ARIs difficult to determine. In a castrate-resistant state, the optimum time frame for maximum upregulation by ARIs may vary between 9–14 days. The effect of PSMA upregulation PSMA radioligand internalization and subsequent on tumor DNA damage and long-term response of these ligands in patients with PCa has yet to be confirmed.

#### 4.3.2. Synergy with Concomitant Agents

Radiosensitizing agents, immune modulators, high-dose vitamin C and polyadenosine-diphosphate-ribose polymerase (PARP) inhibitors may have synergistic effects when combined with PSMA-targeted radioligands [63,74,75,76,77,78,79,80,81,82,83,84,85,86]. The mechanism of action of radiosensitizing agents may be to dysregulate the cell cycle, resulting in oxygenated stress and thereby inducing cytotoxicity and inhibiting DNA repair mechanisms [74]. Dysregulation of the cell cycle causes apoptosis, as the G2 checkpoint bars enter into the mitotic phase if conditions are not met. Immune checkpoint inhibitors such as nivolumab, pembrolizumab and ipilimumab may target apoptosis through dysregulation of the checkpoint and cell cycle [74]. Studies combining immune checkpoint inhibitors to PSMA-targeted radiotherapy, such as pembrolizumab in the PRINCE trial and ipilimumab and nivolumab in the EVOLUTION trial, are pending. Interim analysis of the phase 1 PRINCE trial, where ^177^Lu-PSMA-617 is combined with 200 mg pembrolizumab 3 weekly in patients with mCRPC, showed promising results, with a 50% PSA response rate of 76% (95% CI 59–88) and a 12-month OS of 83% (95% CI 67–92) [87,88].

Oxygenated stress caused by high-dose vitamin C may lead to the formation of free radicals, which in turn may cause cytotoxicity in cancer cells, causing growth arrest and apoptosis [75,76,77,78]. A small clinical study showed that oxygenated stress caused by vitamin C is associated with more DNA damage when combined with ^177^Lu-dotatate in neuroendocrine tumors [79]. A total of 34 patients treated with ^177^Lu-PSMA for PCa (n = 20) or ^177^Lu-dotatate for neuroendocrine tumors (n = 14) were stratified over 27 control cycles without vitamin C and 24 cycles with 1500 mg QD vitamin C, 2 days prior to ^177^Lu-treatment. The serum levels of MDA, a biomarker of the effect of free radicals on the cell membrane, were significantly elevated in the control cycles, confirming the oxidative stress caused by ^177^Lu [79]. Measurements of glutathione and glutathione reductase, markers for oxidative stress, were not significantly different, which might be caused by the small number of patients [79]. In conclusion, vitamin C as a radiosensitizer is an interesting therapeutic option but should be explored further.

Another pathway that may cause synergistic effects in PSMA-targeted radioligands is the PI3K-AKT pathway. The PI3K-AKT pathway is upregulated in 30–60% of PCa, especially in PCa with a high Gleason score (>8) and in CRPC [80]. The PI3K-AKT pathway crosstalks to the androgen receptor pathway (Figure 3) [63,80,81,82]. The PI3K and androgen pathways may regulate each other by a negative feedback loop [80]. Inhibition of PI3K caused growth arrest and activated the AR pathway, measured by increased mRNA levels [81]. This crosstalk was also observed in breast cancer, where inhibition of PI3K led to an increase in HER3, a pathway with similar crosstalk compared to PI3K-AR [83]. Besides the crosstalk between PI3K and AR, the inhibition of AR also leads to increased activity of PI3K. Theoretically, the negative regulation of AR to PI3K is established through stabilization of the AKT phosphatase PHLPP (PH domain leucine-rich repeat protein phosphatase) [81]. In vitro, the pharmacological inhibition of the AR pathway and PI3K pathway resulted in a potentially synergistic tumor reduction of 84% [81]. This was confirmed in LNCaP xenografts, where PI3K inhibition plus AR inhibition led to a reduction of 24% in tumor size, compared to a single inhibition pathway or vehicle of 30–31% or 28% increase in tumor size, respectively [81]. This in vitro and in vivo work was also performed by Thomas et al., where similar results were found, namely an induction of apoptosis, delay in tumor progression and an activation of the AR pathway after inhibition of PI3K [80]. When the PI3K inhibitor is combined with the ARI bicalutamide, inhibition of the cell cycle is found, resulting in an induction in apoptosis. The inhibition of both pathways may cause a delay in CRPC progression of the tumor in LNCaP xenograft mice [80]. A randomized phase Ib/II trial of enzalutamide with or without a PI3K/mTOR inhibitor in patients with mCRPC observed a delay in radiographic PFS from 2.9 to 3.7 months (*p* = 0.069) [89]. The combination of AR plus PI3K pathway inhibitors with RLT has not yet been investigated.

Another interesting group of potentially synergistic drugs are the PARP inhibitors. Physiologically, PARP recognizes single-strand (ss)-DNA breaks and initiates the recruitment of DNA repair proteins, thereby facilitating DNA repair. PARP inhibitors facilitate the conversion of ss-DNA breaks into double-strand (ds)-DNA breaks. A significant increase in PARP on imaging was shown after treatment with RLT in xenograft mice, compared to control animals [90]. Combining PARP inhibitors with RLT should theoretically increase cytotoxicity by inhibiting the repair mechanism [84]. However, PARP inhibitors did not significantly contribute to less tumor growth or better survival in PC3-PIP xenograft mice [85]. Clinical trials in humans combining ^177^Lu-PSMA with PARP-inhibitors are ongoing (NCT03874884, Lu-PARP trial). Prasad et al., described a case report of a 71-year-old man treated with ^177^Lu-PSMA and the PARP-inhibitor olaparib, who showed good tolerance and reasonable PSA decline [86].

Finally, there are indications that glucocorticoid receptor induction leads to a decreased expression and inhibition of the activity of MAP kinases and other transcription factors such as NF-κB, p53 and STAT1, associating the glucocorticoid receptor with tumor suppression in the prostate [63,74,91]. However, no significant improvement in response rates on ^177^Lu-PMSA (*p* = 0.48), no improvement in PFS (*p* = 1.0) and no differences in adverse events were observed in patients (n = 40) treated with dexamethasone 4 mg QD for the first five days compared to patients (n = 31) who did not receive dexamethasone in the first days of a cycle with ^177^Lu-PSMA [91]. These findings were corroborated by a study of Murga et al., who found nonsignificant antagonism by prednisone and prednisolone compared to no use of corticosteroids [63]. In conclusion, combining glucocorticoid receptor induction with ^177^Lu-PMSA did not lead to the desired result and has no need for further investigation.

### 4.4. Dosing of PSMA-Based Radioligands

To date, more than 3000 patients worldwide have received one or more treatment cycles of ^177^Lu-labeled radioconjugates as targeted therapy for (m)CRPC. A meta-analysis of all doses used in published patient data and their respective PSA outcomes (the most consistently published response parameter) and survival data are shown in Table 1. The table shows that reported doses, number of treatment cycles, interval and follow-up period vary widely between studies. Almost all studies were performed on patients with (m)CRPC, treated with second-line ARIs, who tolerated chemotherapy. The mean age of patients was comparable among studies.

#### 4.4.1. Dose–Response of ^177^Lu-PSMA Radioligands

Doses of ^177^Lu-PSMA-617 or ^177^Lu-PSMA-I&T range from 2 to 9.3 GBq with intervals up to 12 weeks, or 22.2 GBq in 2 fractionated doses 2 weeks apart. In these studies, single injections, fractionated dosing or dose escalation were used. These differences hamper a direct interpretation of the dose–response relationship of RLT. In the overall data shown in Table 1, no direct association of (cumulative) dose on OS/PFS can be observed. However, within small individual studies, associations between absolute dose and response were observed. An association has been described between cumulative dose and survival [101]. Twenty-two patients who received >14.8 GBq of ^177^Lu-PSMA (either 617 or I&T) showed a better radiographic PFS compared to 23 patients who received a lower cumulative activity of ^177^Lu-PSMA (*p* = 0.03) [101]. A cumulative activity associated with a longer OS in multiple studies [116,127,127]. Cumulative ^177^Lu-PSMA activity ≥18.8 GBq increased the median OS from 49 to 59 weeks (*p* = 0.04) [121], ≥12.95 GBq increased median OS from 33 to 101 weeks (HR 0.04, *p* = 0.001) [131] and >15 GBq increased OS with a mean activity of 6.8 GBq per cycle (*p* = 0.001) [132]. As longer survival results in higher cumulative doses due to a larger number of treatment cycles, these retrospective findings should be explored further in prospective, randomized clinical trials.

Dose escalation studies were also described in the literature. A total of 40 patients with advanced PCa and positive PSMA uptake were treated in 4 escalating dose cohorts of 4 GBq, 6 GBq, 7.4 GBq or 9.3 GBq ^177^Lu-PSMA every 8 weeks for 3 cycles [133]. No major differences in PSA response were seen after the first nor the third treatment cycle, with a PSA response ≥50% observed after the first dose in 40%, 30%, 50% and 30% per respective dosing group [133]. However, the cohorts were too small for group comparisons, and the scope of the study was not to draw any conclusions on optimal dosage strategy but to establish a standard procedure for RLT without performing individualized dosimetry measurements in advance [133]. The advantage of multiple treatment cycles with ^177^Lu-PSMA is shown, adding a PSA response of 23.1% and 19.2% after a second and third cycle, respectively [90].

#### 4.4.2. Interval of ^177^Lu-PSMA Radioligands

In prior studies, intervals ranged between 4 to 12 weeks, with an exception for the fractionated dosing schedule two weeks apart. The number of cycles varied between 1 and 8 cycles (Table 1). No direct association of intervals on OS/PFS can be observed from Table 1. No study specifically compared multiple dosing intervals of ^177^Lu-PSMA radioligands.

Therefore, it is unclear whether intervals beyond one month between ^177^Lu-PSMA treatment cycles can be justified, given a relatively short time span of cell cycle and DNA repair rates.

Theoretically, the dosing interval should align with tumor homeostasis. The cell cycle of healthy cells takes about 24 h and is potentially shorter in tumor cells, given the short G0-phase and quick duplication rate [134]. During the whole cell cycle, but especially during the M-phase (the phase where the cell divides), cells are sensitive to outside influences, for example, radiation [134]. The radiation exposure by ^177^Lu-PSMA radioligands is relatively long, due to the 6.7-day half-life of the radionuclide. In this time frame, it is likely that all tumor cells go through multiple cell cycles.

Ionizing radiation can cause single-strand (ss) and double-strand (ds) DNA damage, which can be repaired by various DNA repair mechanisms. The median time to repair cells ranges between 10 and 126 min, depending on the complexity of the damage and subsequent repair mechanisms [135,136]. Radiation damage by alpha particles, who mainly cause ds-DNA breaks, may take more time to repair compared to damage by beta or gamma rays, which mainly cause ss-DNA damage [135]. The time needed to repair ds-DNA breaks from radiation and the chemotherapeutic cisplatin is similar. In 59 healthy mice, in the first 2 h after a single cisplatin injection, a relatively rapid repair rate was observed, which then proceeded more slowly for approximately 2 days [136]. After these 2 days, more than 70% of the reparable DNA damage was repaired. The remaining reparable DNA damage was repaired at a relatively slow rate until it reached the detection limit around 30 days [136]. This repair rate advocates for a shorter treatment interval than the 6–12 weeks in most current studies.

In addition to variability in administered ^177^Lu-PSMA doses and dosing intervals, the large variety in patient characteristics may explain some of the variability in outcomes among patients and may influence the dose–response relationship. Previous treatments, stage of disease, sites and number of metastases, age and frailty of the patient may all affect the response to treatment. No studies adjusted the dose based on PSA expression, imaging results or number of metastatic sites. Potentially, tumors with a higher PSA expression do not fully saturate their targets and may benefit from higher doses.

Summarizing, much is unknown about the influence of dose, intervals, cumulative dose and patients’ characteristics on the response of ^177^Lu-PSMA radioligands in PCa. Variation in dosing schedules and intervals makes a direct comparison and recommendations for the optimal dosing strategy difficult. It would be interesting to study alternative treatment regimens for ^177^Lu-PSMA radioligands with a sufficiently high cumulative dose and a shorter interval.

### 4.5. Adverse Events

The most common adverse events of RLT are listed in Table 2. These adverse events are predominantly caused by off-target radiation on PSMA-expressing organs, such as kidneys and salivary glands and by off-target radiation on non-PSMA-expressing organs with a low maximum tolerability for radiation, for example, the red marrow. Less frequently observed adverse events are diarrhea, constipation, weight loss, mucositis, stomatitis, dysgeusia and pain flares [10,13,18,23,24,57,91,93,94,96,99,108,109,110,116,118,119,120,126,128,130,137,138,139,140,141]. Note that most adverse events have been extracted from retrospective studies, where it is not clear whether the adverse events were related to the treatment or to progression of disease.

Xerostomia is most likely caused by the radiation dose of the salivary and lacrimal glands. The tissues of the salivary glands, including the adipose tissue and acinar cells, produce saliva. After treatment with internal radioligands, it is mostly the acinar cells that are compromised, as shown in histopathological studies [34]. This causes a reduction in volume of the salivary glands and a decreased saliva flow [34]. The damage to the salivary and lacrimal glands is predominantly reversible, but irreversible in some after administration of alpha emitters. Reversible xerostomia is resolved in about 3 to 4 weeks [142,143]. The dose dependence of xerostomia is unclear, and studies are conflicting [126,137,144]. The percentage of patients with xerostomia and the severity of xerostomia increase with an increasing number of RLT sessions [137].

Hematological adverse events, among which thrombocytopenia, leukopenia and anemia are most common, are mostly reversible. A dip in blood count is seen around day 30, where after recovery will start [138,145]. Mostly, these variations in blood count during the weeks after treatment with RLT are not clinically significant [23]. As stated earlier, 79% of hematological toxicity co-occurred with tumor progression and was not directly related to therapy [138]. Grade III and IV toxicity mostly occurred after multiple cycles of treatment [138]. Dose dependency of hematological toxicity has not been shown. Afshar-Oromieh et al., compared hematological toxicity in three dose groups of <3.5, 3.5–5.0 and >5.0 GBq and found no association between dose and decline of platelets and leukocytes [137].

Nephrotoxicity is one of the main concerns in RLT and is based on the expression of PSMA on renal tubules. In a clinical cohort study of 22 patients treated with ^177^Lu-PSMA-617, 15 had an eGFR between 40 and 60 mL/min and 7 patients below 40 mL/min at baseline. All patients received two, four or six cycles of ^177^Lu-PSMA-617 with a mean cumulative dose of 40.1 GBq [146]. Surprisingly, in almost all patients, the eGFR improved significantly from 45.0 ± 10.7 mL/min to 52.0 ± 17.4 mL/min after two cycles, 55.6 ± 18.8 mL/min after four cycles and 44.3 ± 9.3 mL/min after six cycles of ^177^Lu-PSMA-617 [146]. Moreover, no significant correlation between dose and change in eGFR was found. As renal function deterioration did not occur, even in this cohort of patients with a compromised baseline renal function, the concern of nephrotoxicity in RLT may be overestimated [129,146,147,148].

#### Management of Adverse Events

Various strategies have been investigated for the management of adverse events. Ice packs, Botox injections and glutamate tablets have been used to protect the salivary glands. Mannitol infusion as an osmotic agent and PSMA inhibitor 2-PMPA were used to protect from renal toxicity. Spaglumic acid eye drops were applied to prevent dry eyes [21,29,74,100,145,149,150,151,152]. These studies concerning the management of adverse events showed either nonsignificant results or a significant decrease in side effects combined with a decrease in efficacy. Since the adverse events are generally mild and reversible, randomized trials are not the highest priority. Therefore, the most beneficial strategy to manage adverse events has yet to be found.

## 5. Optimization Strategy and Future Research

As described above, there are several options for further optimization of RLT, where more research is needed to explore the results of these optimization strategies.

First, dosing amount and frequency should be optimized. A cumulative dose of more than approximately 15 GBq, respectively, could lead to a significantly better OS [101,121,131,132]. Therefore, a dosing strategy should be established, in which this high cumulative dose is exceeded in a short period of time. For full optimization of the dosing regimen, the interval could be shortened to prevent the DNA to fully be repaired and damage the DNA before the repair time has exceeded. In conclusion, a dosing strategy with a short interval, high individualized doses and fast exceedance of cumulative dose could lead to an improvement in OS.

Secondly, dose individualization should be performed. A fixed dose of ^177^Lu-PSMA RLT does not lead to an optimal response for every patient due to variability in PK. Therefore, individualization of treatment, in which tumor volume, PSMA density and flow are considered, could lead to better efficacy and less toxicity [35,39]. These doses could be derived using simulations of the physiologically based pharmacokinetic models by Begum and colleagues, after validation of their predictive values [37,38]. The results of dose individualization should then be compared to the fixed-dose strategy.

Lastly, synergetic effects should be used deliberately to improve the efficacy of RLT, for example, the use of upregulation of PSMA by a short course of enzalutamide or other ARIs. As enzalutamide upregulation is a flare phenomenon, a short course of 9 to 14 days of enzalutamide should maximize upregulation to around 30–45%, thereby enabling more target receptors for PSMA-based RLT [59,65,71].

Summarizing, PSMA-based RLT could be further optimized by an alternative dosing strategy, individualization of treatment and deliberate use of synergistic agents.

## Figures and Tables

**Figure 1 biomedicines-10-03020-f001:**
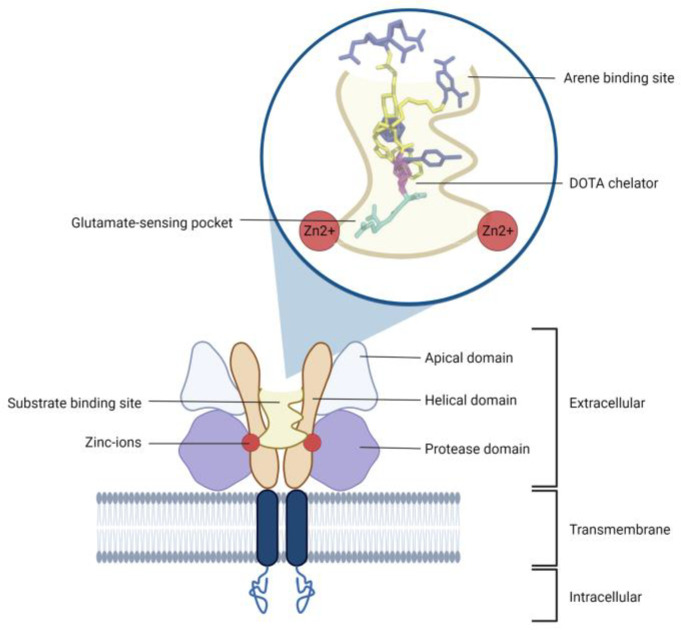
Schematic overview of the structure of a PSMA receptor. DOTA = 1,4,7,10-tetraazacyclododecane-1,4,7,10-tetraacetic acid; PSMA = prostate-specific membrane antigen.

**Figure 2 biomedicines-10-03020-f002:**
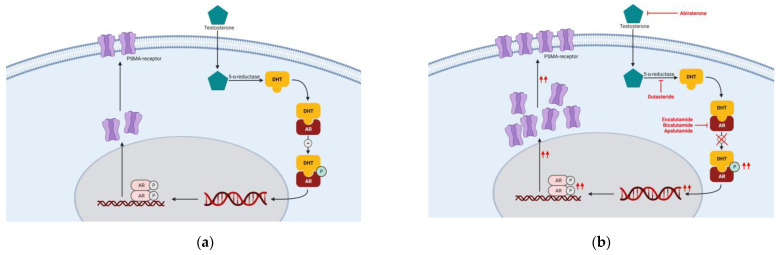
(**a**) Physiological pathway of androgens and regulation of PSMA receptors. DHT coupled to the androgen receptor has an inhibiting effect on the production of PSMA receptors; (**b**) Androgen blockage, for example, by enzalutamide, causes an upregulation of the PSMA receptor due to the loss of the inhibiting effect. DHT = dihydrotestosterone; AR = androgen receptor; PSMA = prostate-specific membrane antigen.

**Figure 3 biomedicines-10-03020-f003:**
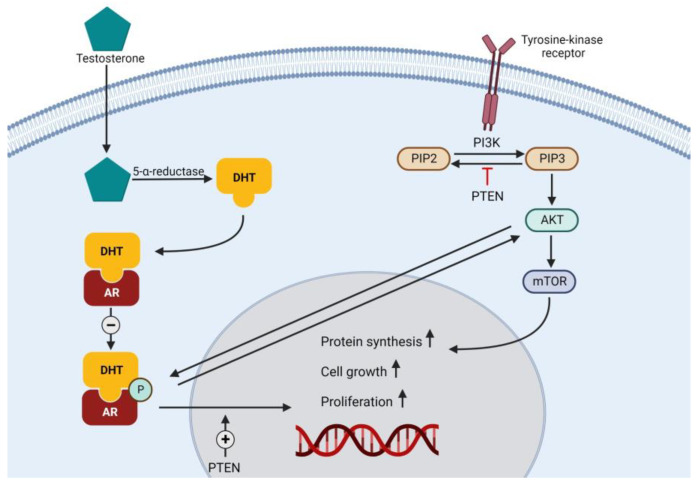
Simplified crosstalk between androgen receptor pathway and PI3K-AKT pathway. DHT = dihydrotestosterone; AR = androgen receptor; PI3K = phosphoinositide-3-kinase; PIP2 = phosphatidylinositol(3,4,5)-biphosphate; PIP3 = phosphatidylinositol(3,4,5)-triphosphate; PTEN = phosphatase and tensin homolog; AKT = protein kinase B; mTOR = mammalian target of rapamycin.

**Table 1 biomedicines-10-03020-t001:** Meta-analysis of all doses using published patient data and their respective PSA outcomes and survival data.

Study Design		Therapy				Patients				PSA Response		Survival		
Type of Study	Follow-Up Period (After Last Cycle)	Agent	Treatment Dose (GBq)	No. of Cycles	Interval (Weeks)	Population	Prior Therapy	No. of Patients	Mean Age	Any Decrease	Decrease ≥50%	Median PFS (Weeks)	Median OS (Weeks)	Ref
P, SS	8 weeks	Lu-PSMA-617	4.4–6.6	1–6		mCRPC	AA, CTx	14	70.57	78.6%	45.4%	-	-	[92]
P, MS	8 weeks	Lu-PSMA-617	5.6	1	n/a	mCRPC	AA, CTx	10	73.5	70%	50%	-	-	[93]
R, SS	8 weeks	Lu-PSMA-617	6	2	12	mCRPC	AA, CTx	24	75.2	79.1%	41.6%	-	-	[94]
P, SS	8 weeks	Lu-PSMA-617	6.0	1–6	8	CRPC	AA, CTx, ^223^Ra	52	70.9	80.8%	44.2%	-	60 (CI 44.2–75.8)	[95]
R, SS, C	Med. 3.4 months	Lu-PSMA-617	6	1–3	8	mCRPC	AA, CTx	49	71.25	67.3%	53.1%	-	-	[96]
P, SS	9 months (range 1–25)	Lu-PSMA-617	3.7–7.4	1–4	8	mCRPC	AA, CTx	21	70.3	76%	62%	-	62.7 (CI 42.1–83.3)	[97]
R, SS, C	Med. 10.3 months	Lu-PSMA-I&T + 617	3.6–8.6	1–10	≥6	mCRPC	AA, CTx, ^223^Ra	83 ^1^	69.3	40%	-	26.1 (CI 13.9–38.3)	46.5 (CI 34.3–58.7)	[13]
R, SS, C	Med. 10.3 months	Lu-PSMA-I&T + 617	3.6–8.6	1–10	≥6	mCRPC	AA, CTx, ^223^Ra	84 ^1^	70.8	57%	-	38.3 (CI 30.9–46.1)	117.8 (CI 80.0–155.7)	[13]
P, SS	15 months (range 6–28)	Lu-PSMA-I&T	5.76	1–5	Personalized	mCRPC	AA, CTx, ^223^Ra	56	72	80.4%	58.9%	59.6	NR	[23]
R, SS	24 weeks (range 15–36)	Lu-PSMA-617	6.1	1–7	8	mCRPC	AA, CTx, ^223^Ra	59	72	91%	53%	18.0 (CI 13.6–22.4)	32.0 (CI 21.1–42.9)	[98]
P, SS, C, phase I/II	Med. 16.3 months	Lu-PSMA-617	7.5	1–6	6	mCRPC	AA, CTx	32	69	91%	62.5%	26.5 (CI 12.2–40.0)	74.4 (CI 28.3–117.8)	[99]
R, SS	4 weeks	Lu-PSMA-617	3.6	2	10	mCRPC	AA, CTx	5	68	-	-	-	-	[100]
R, MS, C	26 months (range 18–38)	Lu-PSMA-I&T + 617	>14.8, cum			mCRPC	AA, CTx	45	70	92%	80%	69.6	NR	[101]
P, SS, phase II	-	Lu-PSMA-617	6–8	1–4	6	mCRPC	AA, CTx	14	69.5	71%	36%	-	50 ± 33	[57]
R, SS	8–10 weeks	Lu-PSMA-617	6.0	2	8–10	mCRPC	AA, CTx, ^223^Ra	10	73	85%	60%	-	-	[102]
R, SS	8 weeks	Lu-PSMA-617	4.1–7.1	1	n/a	CRPC	AA, CTx, ^223^Ra	40	71.4	67.5%	32.5%	-	-	[103]
P, SS	8–10 weeks	Lu-PSMA-617	7.4	1–4		mCRPC	AA, CTx	25	69	84%	-	24 (CI 9–52)	-	[104]
R, SS	8–10 weeks	Lu-PSMA-617	3.7–7.4	1	n/a	mCRPC	AA, CTx	10	67.1	70%	-	24.0 (CI 8.0–36.0)	-	[105]
R, SS	8 weeks	Lu-PSMA-I&T	7.4	1–4	8	mCRPC	AA, CTx, ^223^Ra	22	71	81.8%	27.3%	-	-	[106]
R, SS	9.5 months (range 7.0–16.3)	Lu-PSMA-I&T	7.4	1–6	6–8	mCRPC	AA, CTx, ^223^Ra	100	72	-	38%	17.8 (CI 10.4–24.8)	56.1 (CI 43.0–69.1)	[107]
P, SS, phase II	25.0 months (range 12.7–25.2)	Lu-PSMA-617	4–8	1–4	6	mCRPC	AA, CTx	30	71	97%	57%	33.0 (CI 27.4–39.1)	58.7 (CI 45.2–98.7)	[108]
P, MS, C, phase II	6 months	Lu-PSMA-617	6.0–8.5	1–6	6	mCRPC	AA	200	71.6	89.8%	66%	22.2 (CI 14.8–24.8)	NR	[109]
R, SS	7 days	Lu-PSMA-617	5.52	1	n/a	mCRPC	AA, CTx	9	69.0	-	-	-	-	[32]
R, SS	48 h	Lu-PSMA-617	7.2	1–4		mCRPC	AA, CTx	20	65	-	-	-	-	[79]
R, SS	13.7 months (range 9.8–32.3)	Lu-PSMA-617	6	1–6	6	mCRPC	AA, CTx, ^223^Ra	30	70	-	57%	-	49.1 (CI 6.1–140.4)	[110]
R, SS	17 months	Lu-PSMA-I&T	5.5	1–4	8	mCRPC	AA, CTx	20	71	40%	-	-	-	[111]
R, SS	-	Lu-PSMA-617	6.2	1–9	6–8	mCRPC	AA, CTx, ^223^Ra	109	72	70%	32%	-	43.0 (CI 31.3–54.4)	[112]
R, SS	22 weeks (range 14–63)	Lu-PSMA-617	6.9	1	n/a	mCRPC	AA, CTx, ^223^Ra	20	72	90%	65%	19 (CI 12–26)	49 (CI 4–92)	[113]
R, SS	-	Lu-PSMA-617	7.4, cum	2	12	mCRPC	AA, CTx	1	-	100%	100%	-	-	[114]
R, SS	24 weeks	Lu-PSMA-617	4–6	3	8	CRPC	AA, CTx, ^223^Ra	30	71.9	70%	43.3%	-	-	[18]
R, SS	Med. 19.0 months	Lu-PSMA-I&T	6.0	1–7		mCRPC	AA, CTx, ^223^Ra	119	71	76.3%	57.5%	46.5	NR	[31]
P, SS	Until death	Lu-PSMA-617	6	1–6	6–10	mCRPC	AA, CTx, ^223^Ra	32	71.4	71.9%	37.5%	30.4	52.2	[115]
R, SS	7 months (range 3–14)	Lu-PSMA-I&T + 617	6.1	1–5		mCRPC	AA, CTx	191	70	75%	56%	17.4 (CI 13.0–34.8)	52.2 (CI 21.7–78.3)	[116]
R, SS	-	Lu-PSMA-I&T	7.4	1–4		mCRPC	AA, CTx	18	68	-	-	-	-	[117]
P, SS, phase II	-	Lu-PSMA-617	3.7–5.5	1–2	8–12	mCRPC	AA, CTx	43	73	-	30.8%	-	-	[21]
P, SS	10.6 months (range 8.3–21)	Lu-PSMA-617	3–6	2	8	mHSPC	local	10	67.2	100%	50%	-	-	[24]
R, SS	8 weeks	Lu-PSMA-617	5.9		8	mCRPC	AA, CTx	28	73.4	59%	32%	-	29.4	[118]
R, MS	8 weeks	Lu-PSMA-617	5.9	1	n/a	mCRPC	AA, CTx, ^223^Ra	82	73	64%	31%	-	-	[119]
R, MS	16 weeks (range 2–30)	Lu-PSMA-617	2–8	1–4	8–12	mCRPC	AA, CTx, ^223^Ra	145	73	60%	45%	-	-	[120]
R, MS	Until death	Lu-PSMA-617	6.1	1–8	8	CRPC	AA, CTx, ^223^Ra	104	70	66%	56%	-	56 (CI 50.5–61.5)	[121]
R, SS	24 months (range 6–40)	Lu-PSMA-617	7.3	3	4	mCRPC	AA, CTx, ^223^Ra	54	71.6	79%	58%	-	^2^	[122]
R, SS	8 weeks	Lu-PSMA-617	4.0	1–3	8	mCRPC	AA, CTx, ^223^Ra	10	75.5	90%^1^	40% ^1^	-	-	[123]
R, SS	8 weeks	Lu-PSMA-617	6.0	1–3	8	mCRPC	AA, CTx, ^223^Ra	10	70.5	70% ^1^	30% ^1^	-	-	[123]
R, SS	8 weeks	Lu-PSMA-617	7.4	1–3	8	mCRPC	AA, CTx	10	70.5	70% ^1^	50% ^1^	-	-	[123]
R, SS	8 weeks	Lu-PSMA-617	9.3	1–3	8	mCRPC	AA, CTx, ^223^Ra	10	73.5	80% ^1^	30% ^1^	-	-	[123]
P, SS, phase II	-	Lu-PSMA-617	3.7–5.5	1–4	8–12	mCRPC	AA, CTx	9	68.3	-	-	-	-	[124]
P, MS, phase III	Med. 20.9 months	Lu-PSMA-617	7.4	1–6	6	mCRPC	AA, CTx, ^223^Ra	831	71	71.5%	46.0%	37.8	66.5	[10]
P, SS	Until death	Lu-PSMA-617	6.1	1–3	6–10	mCRPC	AA, CTx, ^223^Ra	10	68	60%	20%	-	-	[125]
R, SS, C	Until death	Lu-PSMA-617	6.1	1–4	6–8	mCRPC	AA, CTx	41	72.7	40.2% ^1^	35.1% ^1^	53.5 ^1^ (CI 44.3–62.6)	55.2 ^1^ (CI 44.3–66.1)	[126]
R, SS, C	Until death	Lu-PSMA-617	7.4	1–4	6–8	mCRPC	AA, CTx	37	68.7	57.8% ^1^	53.7% ^1^	41.3 ^1^ (CI 32.2–50.4)	49.1 ^1^ (CI 33.5–65.2)	[126]
P, SS, phase I	Ongoing	Lu-PSMA-617	7.4–22.2, fr	1	n/a	mCRPC	AA, CTx, ^223^Ra	44	69	-	41%	-	69.6 (CI 47.8–NR)	[127]
P, SS	11 months	Lu-PSMA-617	1.11–5.55	1	n/a	mCRPC	AA, CTx	26	66.3	40.4%	10.6%	-	-	[33]
P, SS	2–3 months	Lu-PSMA-617	3.7–8	1–7	8–12	mCRPC	AA, CTx	90	66.5	62.2%	32.2%	47.8 (CI 39.1–56.5)	60.9 (CI 56.5–69.6)	[128]
R, SS	At least 2 months	Lu-PSMA-617	4.0–7.1	1–6	8	mCRPC	AA, CTx, ^223^Ra	55	72	-	-	-	-	[129]
R, SS	-	Lu-PSMA-617	5.7–8.7	1–4	6	mCRPC	AA, CTx	30	70.5	-	-	-	-	[61]
P, SS, phase II	31.4 months (range 25.1–36.3)	Lu-PSMA-617	7.5	1–4	6	mCRPC	AA, CTx	50	71	-	64%	30.0 (CI 26.1–37.8)	57.8 (CI 45.7–81.3)	[130]

^1^ different treatment groups in 1 study, ^2^ only OS of subgroups. Type of study: R = retrospective, P = prospective, SS = single-center, MS = multicenter, C = comparison with alternative treatment or prior treatments. Population: mCRPC = metastatic castration-resistant prostate cancer, CRPC = castration-resistant prostate cancer, mHSPC = metastatic hormone-sensitive prostate cancer. Pretreatment: AA = androgen receptor inhibitor (enzalutamide, bicalutamide, abiraterone, etc.), CTx = chemotherapy (docetaxel or cabazitaxel), Cum = cumulative dose, fr = fractionated dose, PFS = progression-free survival, OS = overall survival, NR = not reached.

**Table 2 biomedicines-10-03020-t002:** Most common adverse events of PSMA-based radioligand therapy.

Adverse Event	Grade I/II	Grade III/IV
Xerostomia	6–88.2%	1.2% ^1^
Hematological toxicity		
Thrombocytopenia	4.8–20.6%	0.9–27%
Leukopenia	6.9–38.2%	0.9–13%
Anemia	14–98%	1–22%
Fatigue	5.9–70%	3–5.9%
Nausea	12.5–71%	1.3%
Nephrotoxicity	19.5–24%	2–5%

^1^ Only described in ^225^Ac.

## Data Availability

No new data were created or analyzed in this study. Data sharing is not applicable to this article.

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
