# Peer review of "Pharmacological Optimization of PSMA-Based Radioligand Therapy"

_biomedicines, 2022, doi:10.3390/biomedicines10123020_

Round 1
Reviewer 1 Report
Authors should be congratulated for their great work. The topic is interesting and challenging. The use of PSMA ligand therapy represents an innovation in PCa management. The manuscript is well-written and easily readable, the methodology is robust and the figures are particularly clear. The manuscript is suitable for publication.
Reviewer 2 Report
This review presents a review of literature related to PSMA-based radioligand therapy; what is known and how could it be improved. The topic is of interest to clinicians. 617 Lu-PSMA has been recently approved by FDA for advanced CRPC and it shows promise in improving the survival of patients that have very limited effective treatment options available. It is of interest to summarize what is known and where improvements in the treatment regimen can be done to personalize the treatment and provide the best treatment option for each patient.
This review could be significantly improved by better organization/flow of the information and fully up-to-day results inclusion. Moreover, focusing on who published the results (names listed at the beginning of sentences) than on the results themselves and their importance makes it harder to make a more comprehensive picture of the current knowledge.
The part that focuses on potential combination therapies is underdeveloped. For example, PARPi, and GR are introduced as potential targets for combinations, but the publications included do not support these stipulations. The paragraph focusing on AR and PI3K cross talk has no data/information about experiments using the combination of RTL and targeting these pathways.
Other comments:
- The review would benefit from English editing. For example, sentences: “Besides, in some cases no PSA change is found, but on imaging radiographic progression has shown [60].” and “Various strategies have been investigated for the management of adverse events, among which ice packs, Botox injections and glutamate tablets for protection of salivary glands, mannitol infusion as a osmotic agent and PSMA-inhibitor 2-PMPA for kidney pro-tection and spaglumic acid eye drops for prevention of dry eyes [21,29,73,95,140,144-147].”
- It seems that some numbers might be missing from the following sentence: “The upregulation of PSMA found in cells for enzalutamide, abiraterone, bicatulamide and dutasteride was 1.9-7.4-, 1.5-5-, 4.2- and 2.4-4.6-fold respectively [62-67].
- Please check that references 2-4 are correct ones.
- The statement is made that “No limitations on date or language were applied in the search, while it is also stated that “ The timeframe within the databases was from inception to 12th of March 2022.”
